# TGF-β induces ST2 and programs ILC2 development

Li Wang[1,2,6], Jun Tang[1,2,3,6], Xia Yang[1,2,5,6], Peter Zanvit[1,6], Kairong Cui[4], Wai Lim Ku [4], Wenwen Jin[1], Dunfang Zhang [1], Nathan Goldberg[1], Alexander Cain[1], Bing Ni[2], Keji Zhao [4], Yuzhang Wu[2]* & WanJun Chen[1]*

The molecular pathways underlying the development of innate lymphoid cells (ILCs) are mostly unknown. Here we show that TGF-β signaling programs the development of ILC2s from their progenitors. Specifically, the deficiency of TGF-β receptor II in bone marrow progenitors results in inefficient development of ILC2s, but not ILC1s or ILC3s. Mechanistically, TGF-β signaling is required for the generation and maintenance of ILC2 progenitors (ILC2p). In addition, TGF-β upregulates the expression of the IL-33 receptor gene *Il1rl1* (encoding IL-1 receptor-like 1, also known as ST2) in ILC2p and common helper-like innate lymphoid progenitors (CHILP), at least partially through the MEK-dependent pathway. These findings identify a function of TGF-β in the development of ILC2s from their progenitors.

[1] Mucosal Immunology Section, National Institute of Dental and Craniofacial Research (NIDCR), National Institutes of Health (NIH), 30 Convent Drive, Bethesda, MD 20892, USA. [2] Institute of Immunology PLA & Department of Immunology, Third Military Medical University, Chongqing 400038, China. [3] Department of Dermatology, 105th Hospital of PLA, Hefei, China. [4] Center for System Biology, National Heart, Lung and Blood Institute (NHLBI), National Institutes of Health (NIH), 31 Center Drive, Bethesda, 20892 MD, USA. [5] Present address: State Key Laboratory of Trauma, Burns and Combined Injury, Department of Wound Infection and Drug, Daping Hospital, Army Medical University (Third Medical University), Chongqing 400038, China. [6] These authors contributed equally: Li Wang, Jun Tang, Xia Yang, Peter Zanvit. *email: wuyuzhang@yahoo.com; wchen@mail.nih.gov

nnate lymphoid cells (ILCs) belong to a newly identified family of lymphocytes that do not express immune cell lineage surface markers or antigen-specific receptor, but play an important role in protective immunity and regulation of homeostasis and inflammation[1,2]. The dysregulation of ILCs may lead to immune pathologies such as allergy and autoimmune diseases. Based on the expression of signature transcription factors and cytokines, ILCs can be grouped into three subsets. Group 1 ILCs (ILC1s) comprise the prototypical ILC1 and classic natural killer (NK) cells. ILC1s express NKp46, NK1.1, and transcription factor T-bet and produce interferon-γ (IFN-γ)[3]. Group 2 ILCs (ILC2s) express lineage transcription factor GATA3 and produce interleukin 5 (IL-5) and IL-13[4,5]. In mice, ILC2s appear to be the most homogeneous subset so far; they are defined by expression of IL-7 receptor (IL-7R, CD127), Sca-1, CD25, GATA3, and IL-33 receptor ST2 (*Il1rl1*). Group 3 ILCs (ILC3s) comprise the classical lymphoid tissue inducer (LTi) cells plus a subset of non-Lti ILC3s mainly found in intestinal mucosa. ILC3 subsets depend on RORγt for their differentiation and produce IL-17A and IL-22[6–8]. Common lymphoid progenitors (CLPs) in the fetal liver and bone marrow (BM) can generate not only T and B cells, but also all ILC subsets. The common innate lymphoid progenitor (CILP) exists downstream of CLP and gives rise specifically to ILCs. Further downstream is the common helper-like innate lymphoid progenitor (CHILP), a committed ILC precursor for all 'helper' ILC subsets but not for NK cells. The ILC progenitor (ILCP), found downstream of the CHILP, can efficiently generate ILC1s, ILC2s, and some ILC3s, but is poor at generating Lti-like ILC3 subsets[9]. Multiple cytokines have been reported to play a role in regulating the development and function of ILCs[10]. IL-7 signaling is involved in the development, differentiation, maintenance, and function of ILCs except NK cells[11]. IL-12, IL-15, and IL-18 activate ILC1 and NK cells to produce IFN-γ; IL-25, TSLP, and especially IL-33 trigger ILC2s to produce IL-5 and IL-13; IL-23 and IL-1β promote RORγt[+] ILC3s to produce IL-22 and/or IL-17[9].

TGF-β is a pleiotropic cytokine that regulates diverse biological processes during fetal development and cell growth, differentiation, motility, and death. The precise effects of TGF-β depend on the cell lineage, the state of cell differentiation or activation, and the particular cytokine milieu[12,13]. TGF-β exerts a considerable influence on the immune system and has an essential role in maintaining normal immune homeostasis such as regulating the development and differentiation of T cells[14–23]. Recent studies have begun to reveal the role of TGF-β in ILCs. For instance, TGF-β can enhance ILC2 function in the lungs to promote airway hyperactivity[24], guide ILC1 differentiation in the salivary glands[25], impede the conversion of NK cells into ILC1-like cells[26], inhibit the activation and function of NK cells[27,28], and regulate the plasticity of ILC3[29]. TGF-β can be expressed and produced by a wide variety of cells within the BM. However, the role of TGF-β signaling in the development of ILCs from their progenitors in the BM remains unknown.

Here we show that TGF-β signaling is required for the development of ILC2s from their progenitors, but not of ILC1s or ILC3s. Specifically, intrinsic TGF-β signaling deficiency in BM progenitors resulted in defective generation of ILC2s in the periphery. Mechanistically, TGF-β promotes the development and maturation of ILC2s from their progenitors accompanied with an upregulation of ST2, which was at least partially regulated via MEK pathway. In addition, TGF-β is also required for the maintenance of mature ILC2s in both physiological and pathological settings.

## Results

**TGF-β signaling is required for the development of ILC2s.** We first determined the expression of TGF-β receptor I (TβR1) and II (TβR2) mRNAs in all mature ILC subsets in the liver and lamina propria (LP), and their progenitors in the BM of normal C57BL/6 mice (Supplementary Table 1). All ILC subsets and their progenitors expressed *Tgfbr1* and *Tgfbr2* mRNA. It seemed that ILC2 precursors (ILC2p) expressed relatively higher levels of *Tgfbr1* and *Tgfbr2* mRNA among the other progenitors, with mature ILC2s being the highest among the three mature ILC subsets (Supplementary Fig. 1).

To study whether TGF-β signaling affects the development of ILCs from their BM progenitors, we created mixed BM chimeric mice in which CD45.2[+] BM cells from tamoxifen- (*Tgfbr2*[−/−]) or oil-treated (Control) *Tgfbr2*[f/f]ER-Cre[+] mice in equal numbers were injected into lethally irradiated CD45.1[+] hosts along with normal CD45.1[+] BM cells at a 1:1 ratio, respectively (*Tgfbr2*[−/−]/45.1 vs. Control/45.1). Prior to BM cell transfer, we did not find significant difference in the BM composition between 5-days oil-treated and tamoxifen-treated mice (Supplementary Fig. 2). Six weeks later, lymphoid cells from the spleens, guts, lungs, and livers were harvested and counted and found to be similar in all chimeric mice. However, flow cytometry analysis of ILC subsets (the gating strategies shown in Supplementary Figs. 3a, b, and 4b, c) in the LP derived from *Tgfbr2*[−/−]/45.1 chimeras revealed a significant decrease in frequency and total number of CD45.2[+] Lin[−]CD127[+]GATA3[+] ILC2s (Fig. 1a, b). However, frequency, but not total number, of CD45.2[+]Lin[−]CD127[+]RORγt[+] ILC3s significantly increased when compared with ILC3s derived from Control/45.1 chimeras (Supplementary Fig. 4a). Similarly, ILC2s in the lungs (CD45.2[+]Lin[−]CD127[+]Sca-1[+]CD25[+]GATA3[+]) from *Tgfbr2*[−/−]/45.1 chimeras also displayed significantly lower frequency and total number than CD45.2[+] ILC2s from Control/45.1 chimeras (Fig. 1c, d). The frequency and total number of CD45.2[+]Lin[−]CD127[+]Tbet[+] ILC1s in the LP were similar in these two chimeras (Supplementary Fig. 4b). Consistently, no significant changes were observed in the frequency and total number of CD45.2[+]Lin[−]NK1.1[+]NKp46[+] ILC1/NK cells in the liver (Supplementary Fig. 4c). The data suggest a specific defect in ILC2s generation in the absence of TGF-β signaling from BM progenitors.

To rule out the possibility that the defect of ILC2s in the chimeras mentioned above was due to initial difference in the number of BM ILC2 progenitors caused by tamoxifen treatment, we injected CD45.2[+] *Tgfbr2*[f/f]ER-Cre[+] BM cells together with normal CD45.1 BM cells at equal ratios into lethally irradiated CD45.1 mice, and immediately treated these chimeras with tamoxifen or oil for five consecutive days. Six weeks later, mice were harvested and analysis of the ILC subsets revealed significantly reduced frequency and total number of CD45.2[+] ILC2s and an increase of ILC3s, in the LP of tamoxifen-treated chimeras compared to those in oil-treated chimeras (Figs. 1e, f). IL-5 and IL-22 are signature cytokines of ILC2s and ILC3s, respectively[30]. We found that both the frequency and total number of IL-5[+] ILC2s in the LP were significantly decreased in tamoxifen-treated chimeras, whereas IL-22[+] ILC3s displayed an increased frequency but not total number in tamoxifen-treated chimeras (Fig. 1g, h and Supplementary Fig. 4a).

To further validate that TGF-β signaling in ILC precursors plays controlling role in the development of ILCs without bystander effects from other hematopoietic precursors, and to achieve more ILC-restricted TβR2 deletion at a primitive stage, CD45.2[+]Lin[−]CD127[+]Flt3[−]α4β7[+] cells (containing ILC-committed progenitors with restricted lineage potential for all ILCs[31]) sorted from the BM of *Tgfbr2*[f/f]ER-Cre[+] mice were adoptively transferred into lethally irradiated CD45.1[+] hosts along with sufficient CD45.1[+] normal BM cells. The chimeric mice were then treated with tamoxifen or oil for 5 days. ILCs in the LP were then analyzed 6 weeks later. We found that

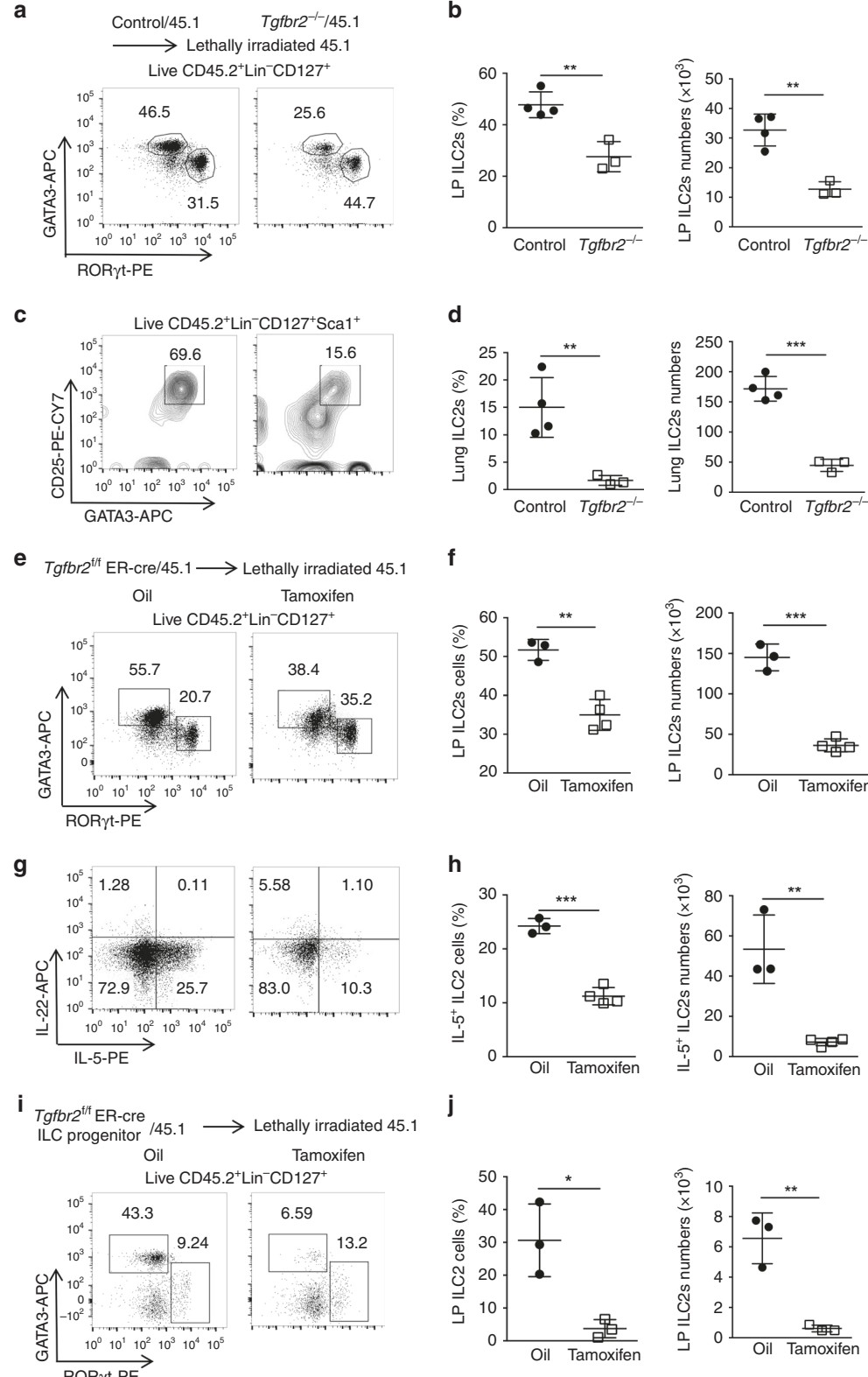

tamoxifen-treated *Tgfbr2*^f/f^ER-Cre+ ILCPs gave rise to hardly detectable ILC2s in the LP of the mixed BM chimeras (Fig. 1i, j). Collectively, these data provide compelling evidence that TGF-β signaling was intrinsically required for the development of ILC2s, but not for ILC1/NKs nor ILC3s. Thus, we focused on the cellular and molecular mechanism of TGF-β signaling in the development of ILC2s.

**Tgfbr2 deletion decreases ILC2p in BM**. ILC2s are developed from an ILC2 lineage-committed precursors (ILC2p) in the BM[32]. ILC2p are developed from CHILP. We next studied whether the inefficient generation of ILC2s in the absence of TGF-β signaling was due to a defective ILC2p in the BM. For this, we analyzed the CHILP and ILC2p cells in the *Tgfbr2*^−/−^/45.1 BM chimeras (gating strategy for ILC2p and CHILP is shown in Supplementary

**Fig. 1 TGF-β signaling is required for the development of ILC2s. a** Representative FACS plots of GATA3$^+$ ILC2s and RORγt$^+$ ILC3s among CD45.2$^+$Lin$^-$CD127$^+$ cells in the lamina propria (LP) of Control/CD45.1 and Tgfbr2$^{-/-}$/CD45.1 BM chimeras reconstituted for 6 weeks. **b** Summary of the frequency and total numbers of CD45.2$^+$ ILC2s in the LP of Control/CD45.1 and Tgfbr2$^{-/-}$/CD45.1 BM chimeras ($n = 3$–4 mice per group). **c** Representative FACS plots of CD45.2$^+$Lin$^-$CD127$^+$Sca1$^+$CD25$^+$GATA3$^+$ ILC2s in the lung of Control/CD45.1 and Tgfbr2$^{-/-}$/CD45.1 BM chimeras reconstituted for 6 weeks. **d** Summary of the frequency and total numbers of CD45.2$^+$ ILC2s in the lung of Control/CD45.1 and Tgfbr2$^{-/-}$/CD45.1 BM chimeras ($n = 3$–4 mice per group). **e** Representative FACS plots of ILC2s and ILC3s in CD45.2$^+$Lin$^-$CD127$^+$ cells in the LP of Tgfbr2$^{f/f}$ER-Cre$^+$/45.1 BM chimeras post-treated with tamoxifen or oil. **f** Summary of the frequency and total number of CD45.2$^+$ ILC2s in the LP of of Tgfbr2$^{f/f}$ER-Cre$^+$/45.1 BM chimeras post-treated with tamoxifen or oil ($n = 3$-4 mice per group). **g** Intracellular staining of IL-5 and IL-22 among CD45.2$^+$Lin$^-$CD127$^+$ cells in the LP of Tgfbr2$^{f/f}$ER-Cre$^+$/45.1 BM chimeras post-treated with tamoxifen or oil. **h** Summary of the frequency and total numbers of IL-5-producing CD45.2$^+$ ILC2s in the LP of Tgfbr2$^{f/f}$ER-Cre$^+$/45.1 BM chimeras post-treated with tamoxifen or oil ($n = 3$–4 mice per group). **i** Representative FACS plots of GATA3$^+$ ILC2s and RORγt$^+$ ILC3s among CD45$^+$Lin$^-$CD127$^+$ cells in the LP of Tgfbr2$^{f/f}$ER-Cre$^+$ ILC progenitor/CD45.1 BM chimeras post-treated with tamoxifen or oil. **j** Summary of the frequency and absolute number of CD45.2$^+$ ILC2s in the LP of Tgfbr2$^{f/f}$ER-Cre$^+$ ILC progenitor/CD45.1 BM chimeras post-treated with tamoxifen or oil ($n = 3$–4 mice per group). Numbers indicate percentages of cells in the indicated gates. All data are representative of at least two independent experiments. Data are shown as mean ± SD; significance was determined by Student's $t$-test (*$p < 0.05$, **$p < 0.01$, ***$p < 0.001$, ns, no significance).

Fig. 3c, e). We found that deficiency of TβR2 did not significantly alter the frequencies of α4β7$^+$Flt3$^-$ progenitor cells and α4β7$^+$Flt3$^-$CD25$^-$ (mainly CHILP) progenitor cells among CD45.2$^+$Lin$^-$CD127$^+$ cells in the BM (Supplementary Fig. 4d). However, there was a significant reduction of CD45.2$^+$ ILC2p cells (CD45.2$^+$Lin$^-$CD127$^+$Flt3$^-$CD25$^+$CD117$^-$ Sca1$^+$GATA3$^+$) derived from Tgfbr2$^{-/-}$ BM in the Tgfbr2$^{-/-}$/CD45.1 chimeras compared to ILC2p cells from Control BM in the Control/CD45.1 chimeras (Fig. 2a, b), although without difference in the expression of GATA3 in the remaining Tgfbr2$^{-/-}$ ILC2p cells (Supplementary Fig. 4e). Consistently, ILC2p cells were also significantly reduced in the BM from tamoxifen-treated mixed BM chimeras created with purified Tgfbr2$^{f/f}$ER-Cre$^+$ ILCPs/45.1 BM cells (Fig. 2c). We did not observe significant difference in ILC2p proliferation between control and Tgfbr2$^{-/-}$ BM chimeras (Supplementary Fig. 4e). Thus, deficiency of TGF-β signaling prevents generation of ILC2p cells in the BM. We hypothetized that the differentiation of ILC2p from CHILP be hampered by deficiency of TGF-β signaling.

***Smad3* deficiency fails to affect the generation of ILC2s.** Next, we studied whether the Smad-mediated canonical pathway is involved in TGF-β controlled development of ILC2s. We focused on the role of Smad3, as it is one of the most important TGF-β downstream receptor-responsive Smads (R-Smads)[33]. We generated mixed *Smad3*$^{-/-}$/45.1 or control/45.1 BM chimeras in which CD45.2$^+$ BM cells from Smad3$^{-/-}$ or littermate control mice were injected into lethally irradiated CD45.1$^+$ hosts together with the same number of wild-type (WT) CD45.1$^+$ BM cells. Unexpectedly, *Smad3*$^{-/-}$/45.1 chimeras displayed comparable frequencies and total number of mature ILC2s in the LP and ILC2p cells in the BM compared to control/45.1 chimeras (Supplementary Fig. 5a, b), indicating that TGF-β programs development of ILC2s through Smad3-independent pathway.

**RNA-seq analysis of *Tgfbr2*$^{-/-}$ ILC2 and ILC2p cells.** The inability of Smad3 deletion to affect ILC2 development encouraged us to study the molecular mechanisms underlying the defect of ILC2s in the absence of TGF-β signaling. We first performed RNA sequencing (RNA-seq) analysis to compare the global transcriptome between freshly isolated Tgfbr2$^{-/-}$ ILC2s from tamoxifen-treated Tgfbr2$^{f/f}$ER-Cre$^+$ mice and WT ILC2s from oil-treated Tgfbr2$^{f/f}$ER-Cre$^+$ mice (Fig. 3a, b). We found that hundreds of genes were changed in Tgfbr2$^{-/-}$ ILC2s compared to WT ILC2s (FDR < 0.1 and fold-change > 1.5). Some ILC2-associated genes including *Gata3*, *Il1rl1* (ST2), *Il-5*, *Atxn1* (Sca1), and *Ccr4*[34] were decreased in Tgfbr2$^{-/-}$ ILC2s, according to RNA-seq analysis (Fig. 3a). In contrast, ILC3-associated genes

such as *Rorc*, *Il22*, *Upp1*, and *Ccl5*[34] were increased in Tgfbr2$^{-/-}$ ILC2s (Fig. 3a). Similarly, signature genes *such as Zbtb16 (PLZF)*, *Ncr1*, *Il2rb*, *Ctla4*, and *Klrb1b* for ILC1/NKs[34] were also upregulated in Tgfbr2$^{-/-}$ ILC2s (Fig. 3a), although there were no significant changes in ILC1/NKs and ILC3s in these knockout mice (Supplementary Fig. 4a, b, c).

However, previously reported ILC2-related transcription factors including *Gfi1*[35], *Rora*[36], *Bcl11b*[37], *Ets1*[38], and *Tcf7*[39] did not show significant difference in gene expression in Tgfbr2$^{-/-}$ ILC2s compared with WT ILC2s. *Sox4*, a downstream target of TGF-β that inhibits GATA-3-induced ILC2s[24], did not show any changes in Tgfbr2$^{-/-}$ ILC2s. *Id2*[40] and *Nfil3*[41], which are critical for all ILC subset development, remained unchanged in Tgfbr2$^{-/-}$ ILC2s. On the contrary, *Zbtb16*, *Tbx21*, and *Rorc*, which are key transcription factors for the specification of ILC1/NK and ILC3 cells, respectively, were upregulated in Tgfbr2$^{-/-}$ ILC2s (Fig. 3b). However, despite the changes observed in RNA-seq analysis, validation of ILC2-related genes by quantitative RT-PCR revealed that only *Il1rl1* was significantly decreased in Tgfbr2$^{-/-}$ ILC2s (Fig. 3c).

To provide direct evidence that TGF-β signaling is crucial for ILC2 development, we next studied the global transcriptome of BM ILC2p. Strikingly, RNA-seq analysis of ILC2p cells consistently revealed significant downregulation of *Il1rl1* in Tgfbr2$^{-/-}$ ILC2p cells (Supplementary Fig. 6a, b). This finding was also confirmed by quantitative RT-PCR analysis (Supplementary Fig. 6c). Consistent with the gene profile in ILC2, some ILC3-associated genes including *Il1r2*, *Gda*, *Capg*[34], and ILC1/NK-unique genes such as *Sell*, *Mmp9*, *Mmp8*, *Cxcr2*, and *Itgam*[34] were also upregulated in Tgfbr2$^{-/-}$ ILC2p cells (Supplementary Fig. 6a). Other ILC2-related transcription genes such as *Gata3*, *Ets1*, *Bcl11b*, *Rora*, and *Tcf7* remained unchanged in Tgfbr2$^{-/-}$ ILC2p cells (Supplementary Fig. 6b). The data collectively indicate that the absence of TGF-β signaling restricted the expression of ILC2-associated genes in the precursors, with *Il1rl1* being the most significantly affected one.

**TGF-β upregulates ST2 and generates ILC2 from BM precursors.** Our previous results indicate that deficiency of Tgfbr2 has an impact in the generation of BM ILC2p but not CHILP cells (Fig. 2, Supplementary Fig. 4d) and the expression of *Il1rl1* was most significantly downregulated in Tgfbr2$^{-/-}$ ILC2p cells (Supplementary Fig. 6). Since IL-33/ST2 has been suggested in the ILC2 development from CLP[35,36], we speculated that TGF-β signaling might be required for the development of ILC2 from CHILP, and this process was related to the regulation of ST2. To test this hypothesis, we performed a series of in vitro ILC2 development assays by co-culture of BM progenitors with OP9 stromal cells that express the Notch ligand DL1 (OP9-DL1 cells).

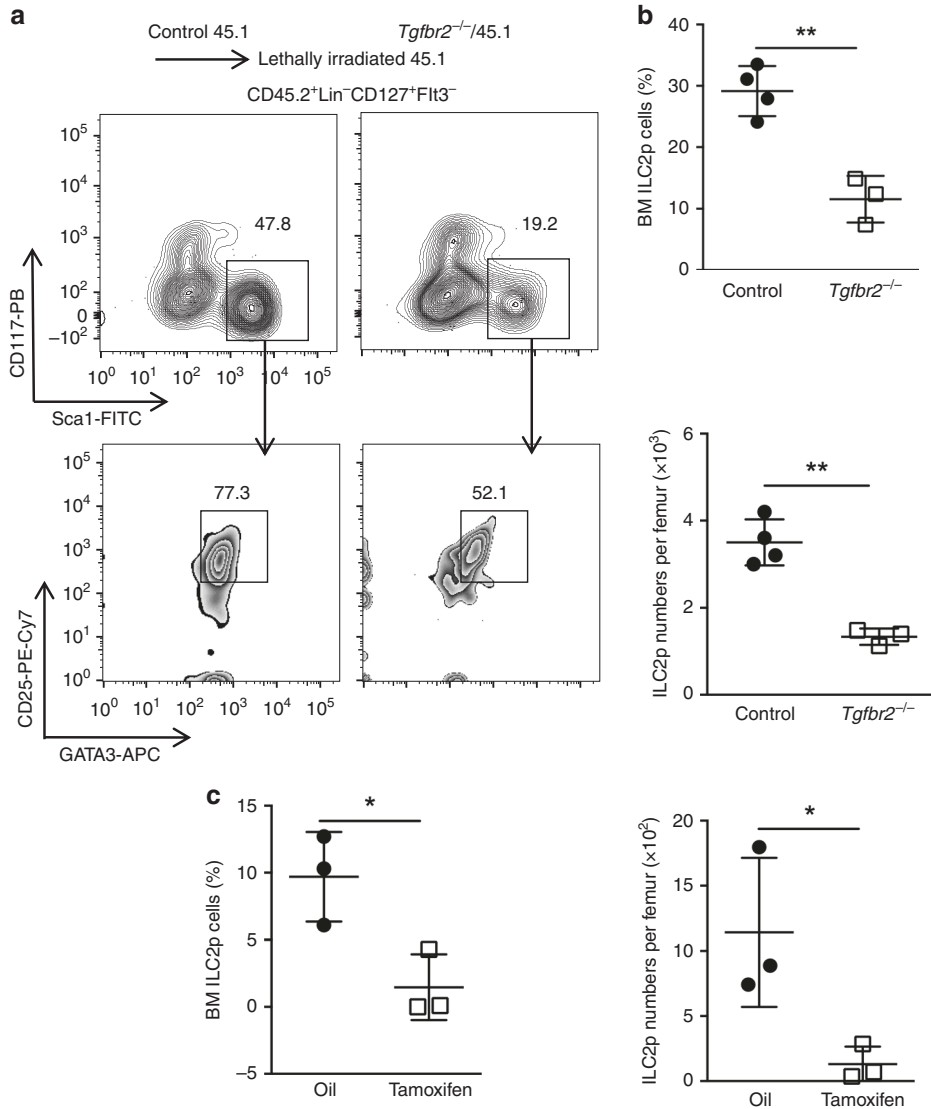

**Fig. 2 TβR2 deletion decreases ILC2 lineage-committed precursor (ILC2p) in the bone marrow. a** Representative FACS plots of CD117⁻Sca1⁺CD25⁺ GATA3⁺ ILC2p gating on CD45.2⁺Lin⁻CD127⁺Flt3⁻ cells in the BM of Control/CD45.1 and *Tgfbr2⁻/⁻*/CD45.1 BM chimeras. **b** Summary of the frequency and total numbers of ILC2p cells in the BM of Control/CD45.1 and *Tgfbr2⁻/⁻*/CD45.1 BM chimeras (*n* = 3–4 mice per group). **c** Summary of the frequency and total numbers of ILC2p cells in the BM of *Tgfbr2*^f/f^ER-Cre⁺ ILC progenitor/CD45.1 BM chimeras post-treated with tamoxifen or oil (*n* = 3–4 mice per group). Numbers indicate percentages of cells in the indicated gates. All data are representative of two independent experiments. Data are shown as mean ± SD; *\*p* < 0.05, *\*\*p* < 0.01 (Student's *t*-test).

Cells were cultured in the ILC2-polarizing medium in the presence of IL-7 and IL-33 cytokines[36]. First, we purified Lin⁻ CD127⁺α₄β₇⁺Flt3⁻CD25⁻ CHILP cells[42] (sorting strategy shown in Supplementary Fig. 3e) from BM of tamoxifen-treated mice (*Tgfbr2⁻/⁻*) or oil-treated control mice (WT). CHILP cells were co-cultured with OP9-DL1 monolayers for 13 days in ILC2-polarizing media with or without TGF-β1. We observed that IL-7 and IL-33 stimulation without exogenous TGF-β1 generated small number of ILC2s defined as CD45.2⁺Lin⁻ CD127⁺GATA3⁺ (containing both ST2⁺ and ST2⁻ cells) from WT CHILP cells, which was substantially reduced with inclusion of TGF-β receptor inhibitor SB431542 in the culture (ST2⁺ ILC2 cells were almost completely eliminated) (Fig. 4a). In contrast, addition of exogenous TGF-β1 into WT CHILP cultures generated much more ILC2 cells and further enhanced their ST2 expression, but only slightly increased GATA3 (Fig. 4a). Notably, among these ILC2 cells, ST2⁺ subpopulation was dominant in both frequency and absolute numbers (Fig. 4a). Consistently,

profoundly lower number of ILC2s (almost all ST2⁻) were generated from *Tgfbr2⁻/⁻* CHILP cultures regardless of whether or not supplement of exogenous TGF-β (Fig. 4a). We obtained similar results using *Tgfbr1⁻/⁻* CHILP cells (Supplementary Fig. 7).

We then investigated the role of TGF-β signaling on BM ILC2p under the same ILC2-polarizing culture conditions, as ILC2p is the immediate precursors of ILC2[32]. Although WT ILC2p cells isolated from BM express high level of ST2 (almost 100%) (Supplementary Fig. 3e), WT ILC2p cells cultured in the presence of IL-7 and IL-33 cytokines for several days showed a reduced expression of ST2. Addition of exogenous TGF-β1 into WT ILC2p cell culture substantially upregulated their ST2 expression and increased ST2⁺ cell subpopulation, without significant changes of GATA3 (Fig. 4b, c). Conversely, blockade of endogenous TGF-β signaling with SB431542 substantially reduced generation of ST2⁺ ILC2 cells as well as the ST2 expression in ILC2p cultures (Fig. 4c). Taken together, these data

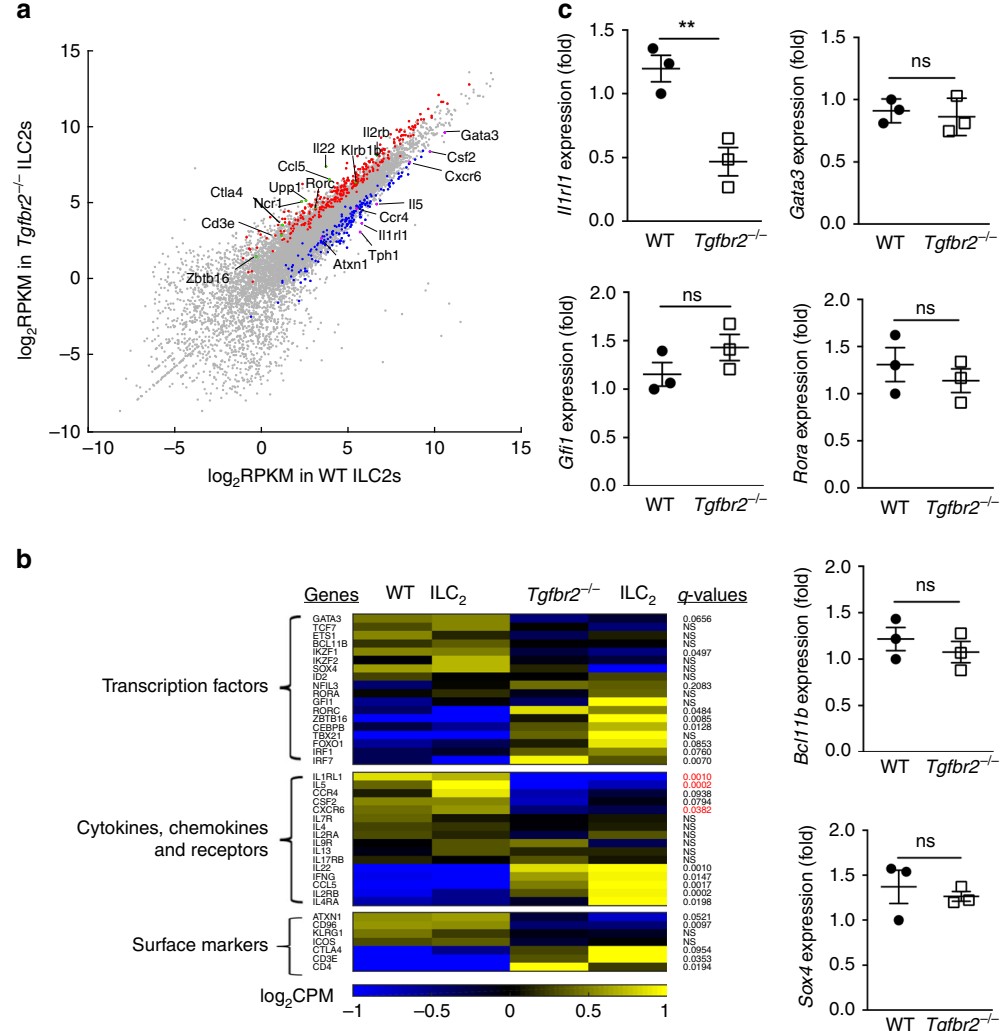

**Fig. 3 TGF-β signaling sustains expression of genes critical for the ILC2 program. a** RNA-seq analysis of sorted ILC2 (CD45.2+Lin−Sca1+CD25+ KLRG1+) in the LP of tamoxifen- (*Tgfbr2−/−*) or oil-treated (Control) *Tgfbr2*f/fER-Cre+ mice. Data are shown as a scatter plot of log2 of FPKM (fragments per kilo base of transcript per million mapped reads) from *Tgfbr2−/−* and WT ILC2s. Blue dots represent downregulated genes, and red dots represent upregulated genes in *Tgfbr2−/−* ILC2s. Genes of interest are indicated and marked in purple (downregulated) and green (upregulated). **b** Heat map of the expression of select genes (left margin) in WT and *Tgfbr2−/−* ILC2s, grouped according to the function of their products (left margin). NS, not significant (*q* value, ≥ 0.2) (Student's *t*-test, Benjamini and Hochberg correction for multiple tests). **c** Quantitative RT-PCR analysis of the indicated genes mRNAs expressed in the purified *Tgfbr2−/−* and WT ILC2s in LP. Data shown are combined data of three independent experiments and are presented as mean ± SEM (\*\**p* < 0.01, ns, no significance).

revealed that TGF-β signaling is essential for the induction of ST2 from CHILP, which safeguard the development of ST2+ ILC2p cells from CHILP progenitors; and TGF-β is also required to prevent downregulation of ST2 on ST2+ ILC2p cells.

**TGF-β increases *Il1rl1* in ILC2 precursors via MEK pathway.** Next, we studied the molecular mechanisms underlying TGF-β-mediated ST2 upregulation in BM CHILP and ILC2p cells. As Smad3-deficiency had no effect on ILC2 development (Supplementary Fig. 5), we determined that TGF-β1 treatment induced a similar (or even stronger) increase in *Il1rl1* mRNA level in *Smad3−/−* CHILP and ILC2p cells compared to their littermate WT control (Fig. 5a), suggesting a Smad3-independent pathway in ST2 upregulation.

We then investigated non-canonical downstream of TGF-β signaling pathways involved in the induction of ST2 in ILC2 progenitors. TGF-β-activated kinase (TAK1) and MEK are two important mediators of the TGF-β-triggered Smad-independent

pathways[33]. We cultured WT CHILP and ILC2p progenitors with OP9-DL1 cells in the ILC2-polarizing conditions in the presence and absence of exogenous TGF-β1 for 24 h. Moreover, we included SB431542, 5z-7-oxozeaenol, or U0126 to block TβR1 signaling, the TAK1-mediated pathway, or MEK-1/2 pathway, respectively. Gene expressions of ILC2-related genes *Gata3, Tcf7, Gfi1, Bcl11b, Rora, Ets1,* and *Il1rl1* were determined using quantitative PCR. Only gene expression of *Il1rl1* was significantly increased in both CHILP and ILC2p cell in response to TGF-β1 treatment (Fig. 5a). *Gata3, Tcf7, Ets1, Gfi1, Bcl11b, and Rora* did not significantly change in ILC2p precursors in response to TGF-β stimulation, although some of them were slightly upregulated in CHILP cells upon TGF-β1 treatment (Supplementary Fig. 8). As expected, inclusion of SB431542 completely abolished TGF-β1-mediated *Il1rl1* mRNA induction in both ILC2p and CHILP cells (Fig. 5a). Blockade of the TAK1-mediated non-canonical pathway with 5z-7oxozeaenol failed to change *Il1rl1* upregulation induced by TGF-β1 (Fig. 5a). Furthermore,

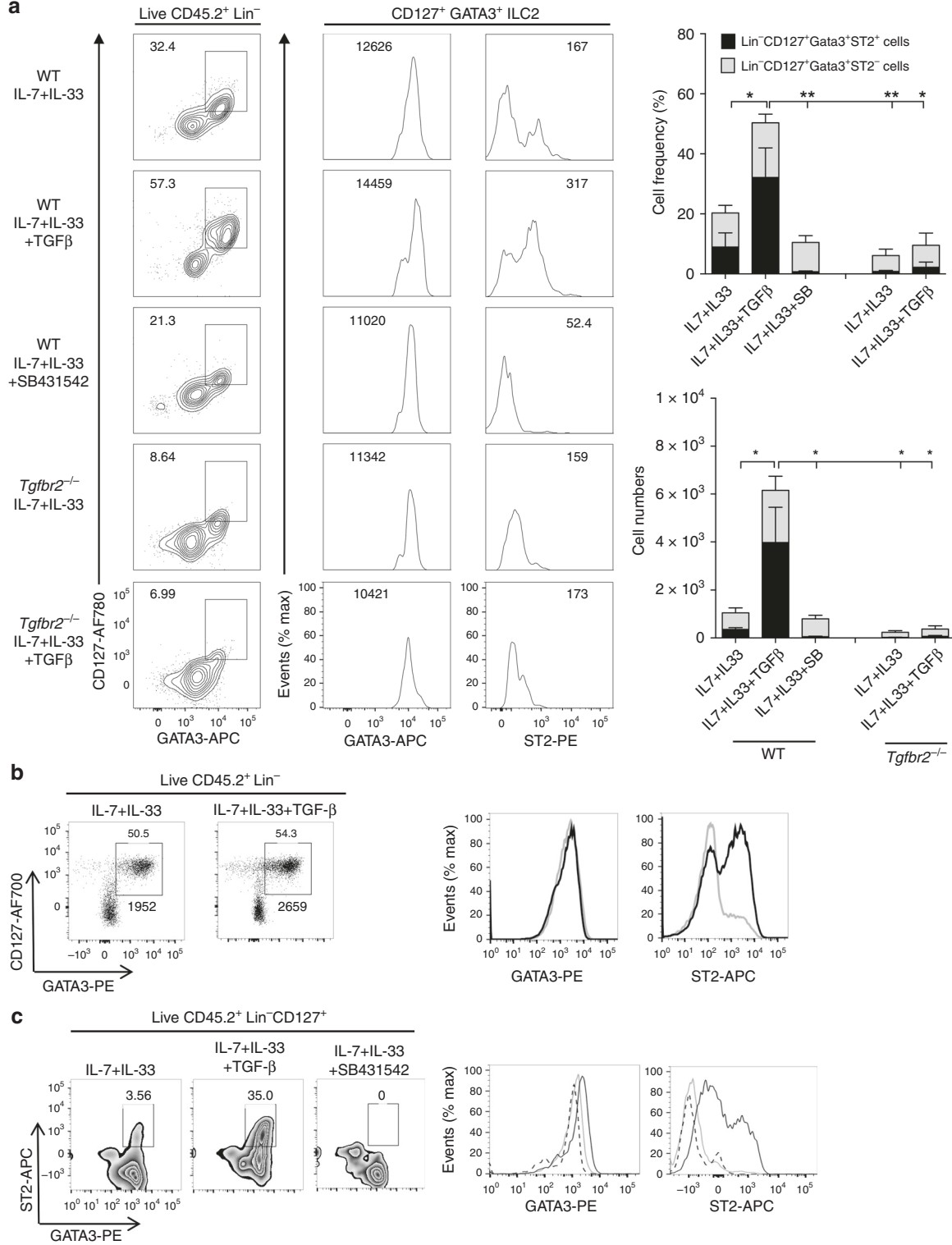

*Tak1*[−/−] ILC2p and CHILP cells exhibited an increase in *Il1rl1* mRNA comparable to that of their WT counterparts in response to TGF-β1 (Fig. 5b). Induction of *Il1rl1* in *Tak1*[−/−] precursors was indeed due to TGF-β signaling, because SB431542 completely abolished TGF-β1-induced *ll1rl1* upregulation (Fig. 5b). Unexpectedly, inhibition of MEK1/2 pathway with U0126 significantly suppressed TGF-β1-induced *Il1rl1* expression in both WT ILC2p and CHILP cells (Fig. 5a), suggesting a role for MEK1/2 mediated pathway in *Il1rl1* upregulation. Importantly, blockade of MEK1/2 in *Smad3*[−/−] ILC2p and CHILP precursors also partially blocked

the TGF-β1-induced *Il1rl1* mRNA increase (Fig. 5a). As it is known that IL-33 is an important cytokine that enhances ST2 expression[43], we next examined whether the TGF-β-mediated increase in ST2 expression was IL-33 independent. Strikingly, TGF-β1-induced *Il1rl1* upregulation in ILC2p and CHILP precursors was not dependent on IL-33, as TGF-β1 was able to induce a similar or even higher increase in *Il1rl1* mRNA in the cultures without exogenous IL-33 (Fig. 5c). Consistently, both the MEK1/2 and TβR1 inhibitors efficiently blocked the TGF-β-mediated increase in *ll1rl1* expression in IL-33-deficient cultures

**Fig. 4 TGF-β promotes in vitro development of ILC2 from BM precursors and upregulates ST2 expression. a** WT or *Tgfbr2*⁻/⁻ CHILP cells (Lin⁻CD127⁺Flt3⁻α₄β₇⁺CD25⁻) were co-cultured with OP9-DL1 cells in presence of IL-7 (20 ng/ml) and IL-33 (20 ng/ml) or TGF-β (2 ng/ml) or TβR1 inhibitor SB431542 (5 μM) for 13 days and analyzed using flow cytometry. Left panel: representative FACS plots depict frequency of ILC2 cells generated from WT and *Tgfbr2*⁻/⁻ CHILP, respectively. Middle two panels: histograms show representative expression of GATA3 and ST2 in WT and *Tgfbr2*⁻/⁻ LC2 cells. Numbers indicate mean fluorescence intensity (MFI). Right upper panel is showing frequency of both Lin⁻CD127⁺GATA3⁺ST2⁺ and Lin⁻CD127⁺GATA3⁺ST2⁻ among CD45⁺ live cells in CHILP cultures (day 13) of WT and *Tgfbr2*⁻/⁻ cells. Right lower panel is showing total numbers of both Lin⁻CD127⁺GATA3⁺ST2⁺ and Lin⁻CD127⁺GATA3⁺ST2⁻ among CD45⁺ live cells in CHILP cultures (day 13) of WT and *Tgfbr2*⁻/⁻ cells. Data shown are pooled of three (*Tgfbr2*⁻/⁻ CHILP) and five (WT CHILP) independent experiments. Statistical analysis was performed using one-way ANOVA followed by Tukey's multiple comparison test (*$p < 0.05$; **$p < 0.01$). **b** WT ILC2 precursors (ILC2p; Lin⁻CD127⁺Flt3⁻α₄β₇⁺CD25⁺) cultured with OP9-DL1 stromal cells with IL-7 and IL-33, in the presence (black solid line) or absence (gray solid line) of TGF-β1, and analyzed for CD127⁺GATA3⁺ ILC2s on day 13 of culture, pre-gated with live CD45.2⁺Lin⁻ cells. The above numbers indicate the frequency and the numbers below indicate the events number of the indicated gate, respectively. The expression levels of GATA3 and ST2 in these cultured ILC2s were assessed. **c** Representative FACS plots of sorted WT ILC2p cells co-cultured with OP9-DL1 stromal cells plus IL-7 and IL-33, in the presence (black solid line) or absence (gray solid line) of exogenous TGF-β, or with addition of SB431542 (black dashed line) for 13 days. Representative FACS plots showing expression of GATA3 and ST2 among live CD45⁺Lin⁻CD127⁺ cells. Data in **b** and **c** are representative of two independent experiments.

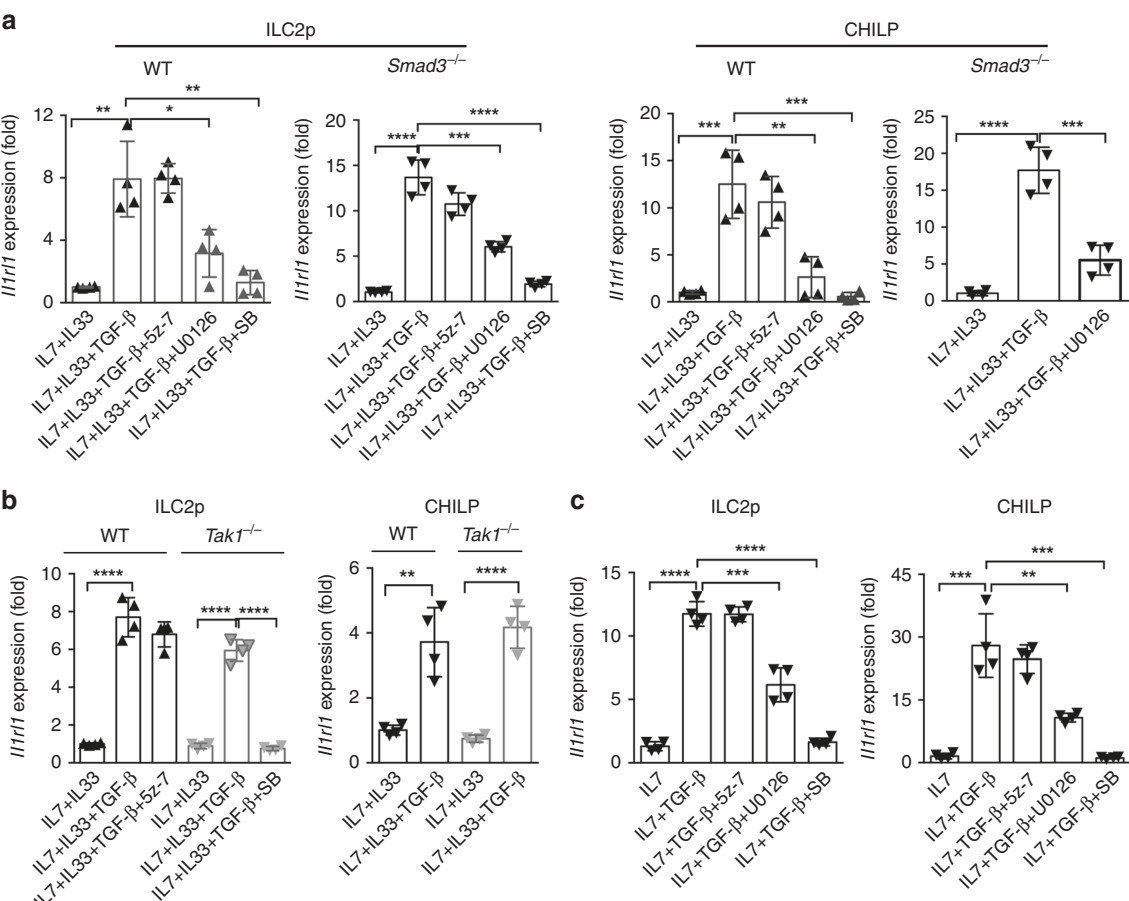

**Fig. 5 TGF-β increases *Il1rl1* mRNA in BM ILC2 precursors partially through MEK-dependent pathway. a** Quantitative RT-PCR analysis of the gene expression of *Il1rl1* in purified ILC2p and CHILP from WT and *Smad3*⁻/⁻ mice cultured in medium containing IL-7 and IL-33, performed 24 h after treatment with TGF-β1 or TGF-β1 and indicated inhibitors, and normalized to *Hprt1* expression. **b** *Il1rl1* mRNA expression in purified WT or TAK1-deficient ILC2p and CHILP cultured in IL-7 and IL-33 containing condition, performed 24 h after treatment with TGF-β1 or TGF-β1 and indicated inhibitors, and normalized to *Hprt1* expression. **c** *Il1rl1* mRNA expression in WT ILC2p and CHILP cultured in medium containing only IL-7, performed 24 h after treatment with TGF-β1 or TGF-β1 and indicated inhibitors, and normalized to *Hprt1* expression. In each experiment, the BM cells were pooled from ten mice in each group before the cultures. Data are pooled from two independent experiments and are presented as mean ± SD. *$p < 0.05$, **$p < 0.01$, ***$p < 0.001$, ****$p < 0.0001$ (Student's *t*-test).

(Fig. 5c). These data collectively indicate that TGF-β upregulates *Il1rl1* expression in both BM ILC2p and CHILP cells at least partially via MEK-dependent, but Smad3- and TAK1-independent, pathway.

**TGF-β increases ST2 and maintains mature ILC2s.** Since IL-33/ST2 signaling is involved in the expansion of mature ILC2s in the periphery[44], we next investigated whether TGF-β signaling plays a role in the homeostasis of mature ILC2s. KLRG1 has been

reported as a specific marker for mature ILC2s[32]. We sorted mature $Tgfbr2^{-/-}$ and WT ILC2s (CD45$^+$Lin$^-$CD127$^+$Sca1$^+$CD25$^+$KLRG1$^+$) (Supplementary Fig. 3d) from the LP of tamoxifen or oil-treated $Tgfbr2^{f/f}$ER-Cre$^+$ mice, respectively, and adoptively transferred these ILC2s into ILC-deficient $Rag2^{-/-}$$Il2rg^{-/-}$ mice[4,5,45] in equal numbers. Twelve days later, the frequencies of ILC2s in the LP of recipient mice were analyzed. TGF-β signaling deficiency resulted in a significant decrease in the frequency and total numbers of mature ILC2s in the LP (Fig. 6a, b). Further analysis revealed that the proliferation of $Tgfbr2^{-/-}$ ILC2s as indicated by Ki67 was reduced compared to WT ILC2s (Fig. 6a, b). Consistently, the proliferation of $Tgfbr2^{-/-}$ ILC2s in the mixed BM chimeras as described in Fig. 1a was lower than in control ILC2s, without changes of apoptosis (Supplementary Fig. 9a, b). The data suggest that TGF-β also maintains the homeostasis of mature ILC2s in the periphery. We then examined CD127 (IL-7R) that is known to be associated with ILC2s homeostasis and maintenance, but found that $Tgfbr2^{-/-}$ ILC2s exhibited normal levels of surface CD127 (Supplementary Fig. 9a). Consistently, RNA-seq analysis showed that $Il7r$ gene expression remained unchanged in $Tgfbr2^{-/-}$ mature ILC2s compared to WT mature ILC2s (Fig. 3b), excluding the involvement of IL-7 signaling in the effect. However, the expression of $ll1rl1$ mRNA was significantly reduced in $Tgfbr2^{-/-}$ ILC2s (Fig. 3c). In addition, flow cytometry analysis confirmed the decrease in ST2 protein in $Tgfbr2^{-/-}$ ILC2s, although these cells expressed normal level of GATA3 (Fig. 6c, Supplementary Fig. 9a). The data suggest that the decrease in $Tgfbr2^{-/-}$ mature ILC2s might be attributable to the decrease of ST2 rather than GATA3 in these knockout cells in the periphery. To provide further evidence of the role of TGF-β1 in mature ILC2s, we cultured mature ILC2s isolated from LP with IL-7 and various concentrations of recombinant TGF-β1, and found that even low concentrations of TGF-β1 (e.g. 0.02–0.2 ng/ml) increased the levels of ST2 protein (Fig. 6d) and $Il1rl1$ mRNA (Fig. 6e). Notably, the same concentrations of TGF-β1 failed to significantly change the levels of GATA3 protein (Fig. 6d) or mRNAs of $Gata3, Bcl11b, Gfi1$, and $Rora$ (Fig. 6e). Thus, these data indicate that TGF-β signaling is important for the maintenance of mature ILC2 cells in the periphery via upregulation of ST2.

**$Tgfbr2^{-/-}$ ILC2s are functionally deficient to HDM-induced airway inflammation**. Although an earlier study showed that epithelial TGF-β enhances ILC2 function in lung through acting as a chemoactive factor for ILC2s[24], the intrinsic impact of TβR2 deficiency on the function of ILC2 cells has not been fully understood yet. $Tgfbr2^{-/-}$ ILC2s in the mixed BM chimeras reduced their ability to produce ILC2 signature cytokines with PMA/Ionomycin restimulation compared with control ILC2s (Fig. 1g, h). Conversely, TGF-β treatment increased the production of IL-5 and IL-13 by ILC2s generated from WT ILC2p cultures in vitro (Supplementary Fig. 10). We then hypothesized that TGF-β signaling also impact the function of ILC2s. To examine this, we sorted WT and $Tgfbr2^{-/-}$ ILC2s and adoptively transferred into $Rag2^{-/-}$$Il2rg^{-/-}$ mice in equal numbers, followed by induction of allergic airway inflammation with house dust mite (HDM)[19] (experimental scheme shown in Fig. 7a). Histological analysis of lungs of the HDM-treated mice revealed a substantial decrease in the inflammatory infiltrates and mucus production in $Rag2^{-/-}$$Il2rg^{-/-}$ mice that received $Tgfbr2^{-/-}$ ILC2s compared to WT ILC2s (Fig. 7b). Moreover, adoptive transfer of $Tgfbr2^{-/-}$ ILC2 significantly decreased the numbers of eosinophils in both the lungs and bronchoalveolar lavage fluids (BALF) in $Rag2^{-/-}$$Il2rg^{-/-}$ mice compared to WT ILC2-

transfered mice (Fig. 7c). Importantly, the frequency and total number of IL-5$^+$ and IL-13$^+$ ILC2s were significantly reduced in the lungs of the HDM-treated $Rag2^{-/-}$$Il2rg^{-/-}$ mice that recieved $Tgfbr2^{-/-}$ ILC2s (Fig. 7d). We were unable to compare the aforementioned factors between $Tgfbr2^{-/-}$ and WT ILC2s in the lungs of unchallenged $Rag2^{-/-}$$Il2rg^{-/-}$ mice due to the insufficient number of cells isolated from the mice. The data altogether indicate that loss of TGF-β signaling results in diminished allergic lung inflammation due to a reduced ability to produce ILC2 cytokines. Thus, TGF-β signaling also impacts the function of ILC2 by regulating their signature cytokines.

## Discussion

The molecular pathways underlying the development of ILCs still remain largely unknown. Here, we revealed a cell-intrinsic role of TGF-β in programming the development of ILC2s from their progenitors, CHILP and ILC2p in the BM. TGF-β is critical for both the upregulation of ST2 and the development of ILC2 from ILC2 progenitors[36], which was at least partially mediated via MEK1/2-dependent, but Smad3- and TAK1-independent, pathways. Our data also indicate that TGF-β signaling impacts the maintenance and function of ILC2s in the periphery.

Several conclusions can be drawn from this study. Firstly, TGF-β signaling plays an intrinsic role in programing ILC2 development; yet is dispensable for ILC1 and ILC3 development. TGF-β is a pleiotropic cytokine that maintains immune homeostasis. For instance, TGF-β inhibits Th2 cell differentiation and cytokine production by repressing expression of GATA3[46]. GATA3 is not only an important Th2 cell regulator, but also a critical transcription factor for ILC2 lineage specification and maintenance. We originally hypothesized that TGF-β signaling might have a negative effect on ILC2 development and differentiation. Strikingly, however, in several physiological BM chimeric models in vivo, we found that TGF-β signaling in BM progenitors intrinsically programs the generation of ILC2s, but not of ILC1s or ILC3s. The deletion of TβR2 selectively lowered ILC2s in periphery and ILC2p in BM, but not the upstream of ILC2p including CHILP, suggesting that the reduction in ILC2s in $Tgfbr2^{-/-}$ chimeras is likely a consequence of defective development of ILC2p from CHILP. CHILP, a committed ILC precursor for all 'helper' ILC subsets, retained normal abilities to produce ILC1s and ILC3s under loss of TGF-β signaling, indicating that TGF-β signaling has minor impact on the development of ILC1s and ILC3s. The similar frequency and number of ILC1s derived from $Tgfbr2^{-/-}$ and control BM chimeras was consistent with another study showing that TβR2 deficiency has minimal impact on the liver and gut ILC1s[25]. The conclusion was also supported by our in vitro ILC2 differentiaiton assays, which clearly demonstrated that TGF-β signaling plays an essential role during differentiation stages from CHILP to ILC2. Moreover, our global transcriptome analysis in ILC2s and ILC2p cells revealed that TGF-β signaling sustains the expression of the ILC2 genetic program and restricts the ILC3/ILC1 genetic program. However, this study does not exclude the role of TGF-β signaling in mature ILC1s and/or ILC3s that reside in certain tissues. For instance, TGF-β has been shown to guide ILC1s differentiation in the salivary glands[25], impede the conversion of NK cells into ILC1-like cells[26], and regulate the plasticity of ILC3s[29].

Secondly, the effect of TGF-β signaling on the development of ILC2s may be at least partially linked to upregulation of ST2 in CHILP and ILC2p cells. So far, several transcription factors including GATA3, RORα, and GFI1 have all been demonstrated in controlling ILC2 cell fate, development, and/or maintenance[32,35,36]. BCL11b was found to maintain ILC2 genetic and functional programs through GFI1, GATA3, and RORα[37,47].

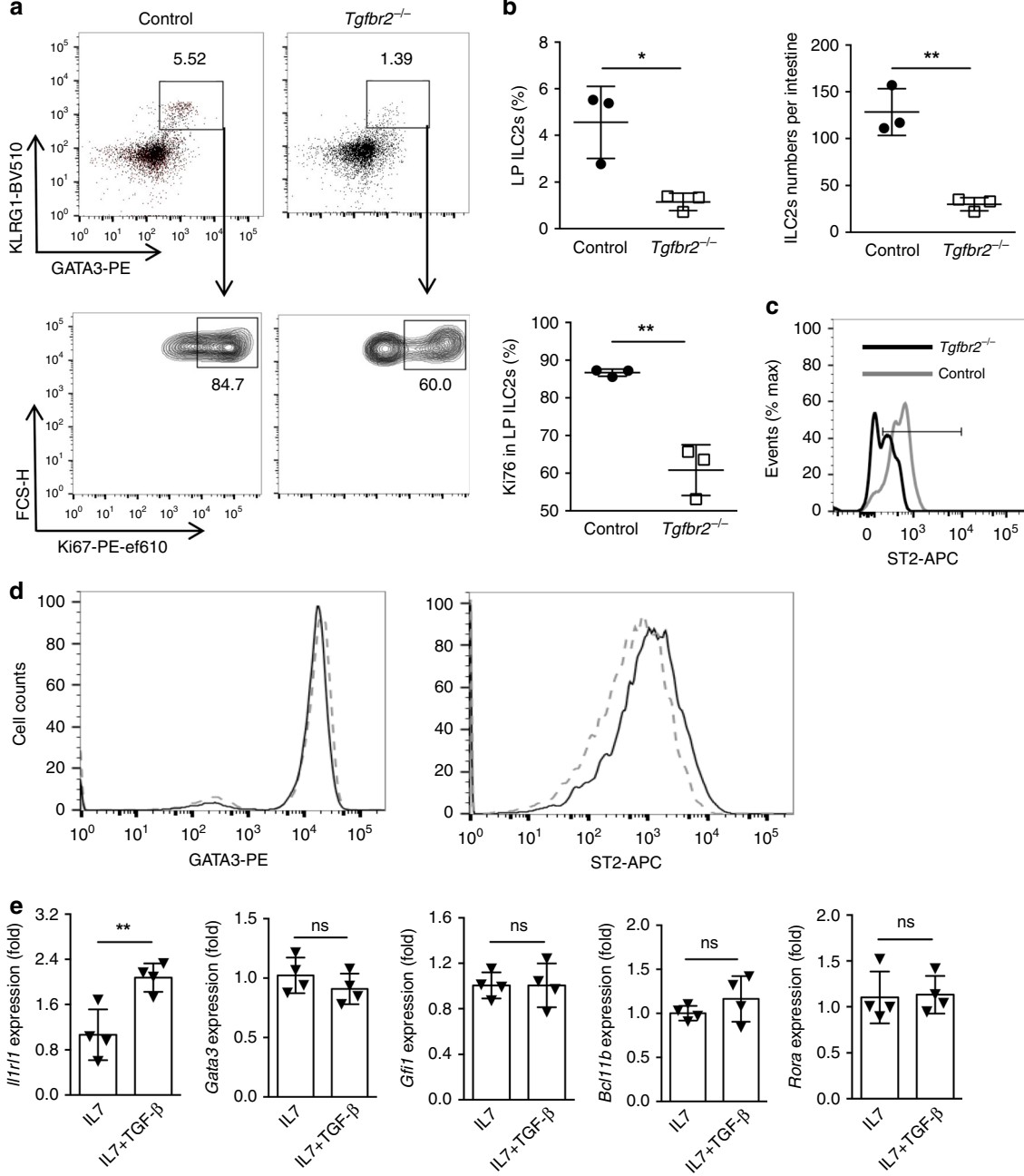

**Fig. 6 TGF-β signaling maintains mature ILC2s in the periphery. a** Sorted mature ILC2s (CD45.2+Lin−CD127+Sca1+CD25+KlRG1+) in the LP of tamoxifen- (*Tgfbr2*−/−) or oil-treated (Control) Tgfbr2^f/f^ER-Cre+ mice were adoptively transferred into *Rag2*−/−*Il2rg*−/− mice. Frequencies and Ki67 expression of GATA3+KLRG1+ ILC2s among CD45.2+Lin−CD127+ cells in the LP of these recipients were assessed by flow cytometry after 12 days. **b** The frequency, total number, and Ki67-positive percentage in LP ILC2s gated in **a**; *n* = 3 mice per group. **c** Representative FACS histograms of ST2 expression by ILC2s in the lung of Control/CD45.1 and *Tgfbr2*−/−/CD45.1 BM chimeras. **d** Representative FACS histograms of sorted WT CD45.2+Lin−CD127+Sca1+ CD25+KlRG1+ ILC2s cultured with IL-7, in the presence (black solid line) or absence (gray dashed line) of TGF-β1 and analyzed for the expression of GATA3 and ST2 among live CD45+Lin−CD127+ cells after 3 days. **e** Quantitative RT-PCR analysis of the relative mRNA expression of selected genes in purified WT ILC2s cultured for 24 h in IL-7 alone or IL-7 plus TGF-β1. In each experiment, sorted LP ILC2 cells were pooled from six mice in each group before cultures. Data are combined of two independent experiments and are presented as mean ± SD. *$p < 0.05$, ns, no significance (Student's *t*-test).

Recently, TCF-1 has been demonstrated to act through both GATA3-dependent and GATA3-independent pathways to promote the generation of ILC2[48]. However, by comparing the global transcriptome among *Tgfbr2*−/− ILC2s or ILC2p cells and their WT counterparts, we did not find significant changes in gene expression of the aforementioned ILC2-associated genes. Further experimental evidence shows that *Tgfbr2*−/−ILC2p and ILC2s expressed a similar degree of GATA3 to their control counterparts at single cell level. As GATA3 has been reported to be critical for the development of all IL-7R+ ILC subsets[49,50] and ILC1s and ILC3s were not affected by the absence of TβR2, we postulate that TGF-β signaling programs the development of ILC2s probably via a GATA3-independent pathway. Several studies have shown that IL-33/ST2 signaling is important for ILC2 development and maturation[35,36,51]. ST2 is known as an IL-33 receptor expressed on Th2, mast cells, ILC2, and ILC2p. ST2 is

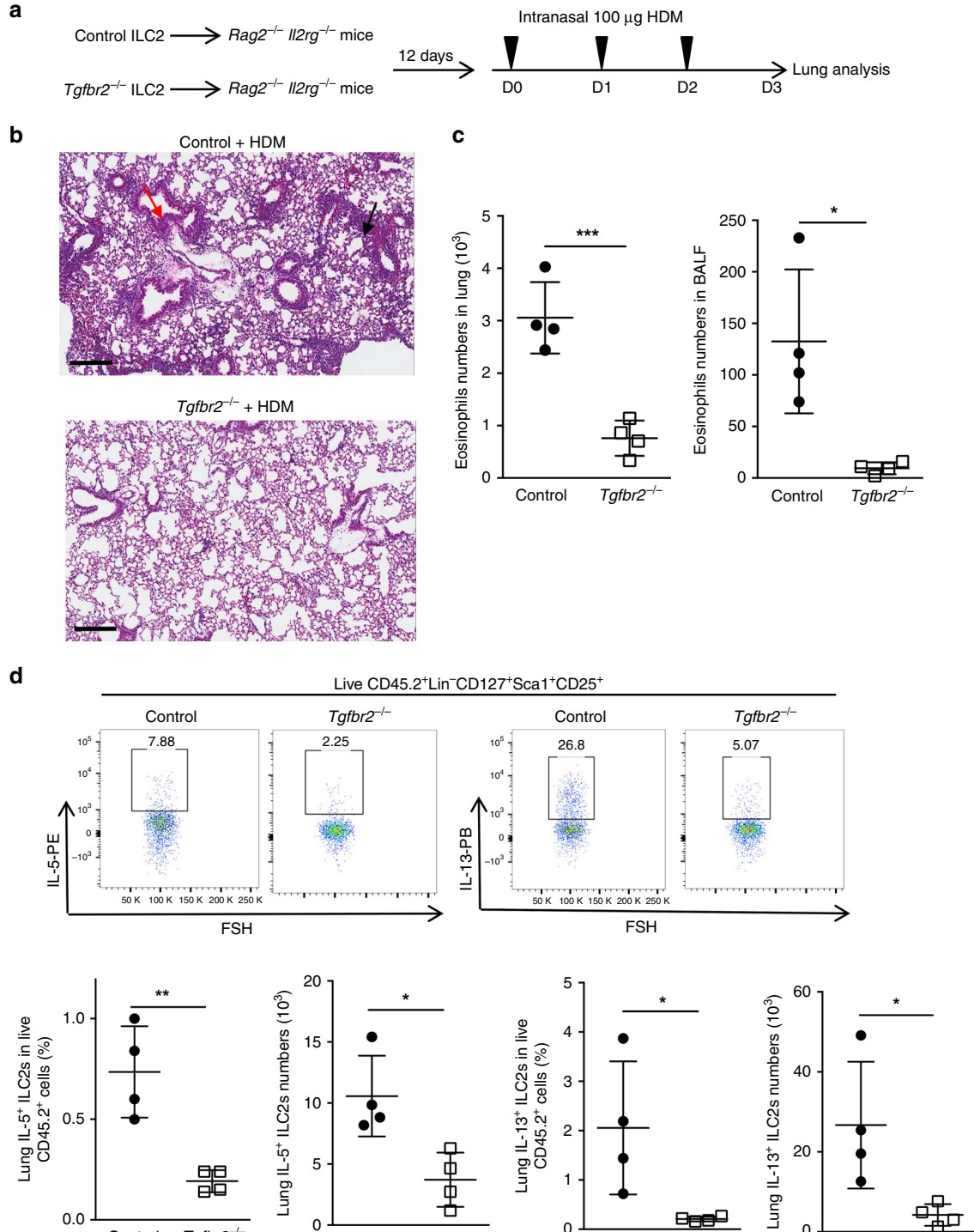

**Fig. 7 TβR2$^{-/-}$ ILC2s are functionally deficient in response to HDM-induced airway inflammation. a** Sorted mature ILC2s of *Tgfbr2*$^{f/f}$ER-Cre$^+$ mice pre-treated with tamoxifen (*Tgfbr2*$^{-/-}$) or oil (Control) were transferred (5 × 10$^5$ per mouse) into *Rag2*$^{-/-}$*Il2rg*$^{-/-}$ mice. Twelve days later, recipient mice were injected intranasally with 100 μg of HDM in 50 μl of PBS on days 0–2 and further euthanized on day 3. BALF was collected and lungs were dissected for analysis. **b** Inflammatory infiltrates (black arrows) and mucus secretion (red arrows) were analyzed by H&E staining of formalin-fixed paraffin-embedded lung sections of HDM treated-recipient *Rag2*$^{-/-}$*Il2rg*$^{-/-}$ mice. Scale bars represent 200 μm. **c** The total numbers of eosinophils in the lung and/or BALF of HDM-treated *Rag2*$^{-/-}$*Il2rg*$^{-/-}$ mice that were transferred with *Tgfbr2*$^{-/-}$ ILC2s and Control ILC2s (*n* = 4 per group). **d** Flow cytometry profiles and summary of the frequency and total numbers of IL5$^+$ ILC2s, IL-13$^+$ ILC2s gated as CD45.2$^+$Lin$^-$CD127$^+$CD25$^+$Sca1$^+$ in the lung of HDM-treated *Rag2*$^{-/-}$*Il2rg*$^{-/-}$ mice that were transferred with *Tgfbr2*$^{-/-}$ ILC2s and Control ILC2s. Numbers indicate the frequency of flow cytometric events (*n* = 4 per group). Data are representative of two independent experiments. Data are presented as mean ± SD; *$p < 0.05$, **$p < 0.01$, ***$p < 0.001$ (Student's *t*-test).

also regarded as an exclusive marker of ILC2p[32]. A defect in the generation of ILC2p is regarded partly as a consequence of an inability to properly induce expression of ST2 in these progenitors[35]. Indeed, our data showed that TGF-β signaling regulates *ll1rl1* gene expression in both ILC2p as well as mature ILC2s. RNA-seq analysis revealed that *Il1rl1* was the most significantly downregulated gene in both *Tgfbr2*$^{-/-}$ ILC2s and ILC2p cells. This result was confirmed by quantitative RT-PCR and flow cytometry analysis. Through a well-recognized in vitro OP9-DL1 co-culture system, we determined that TGF-β treatment promoted the generation of ILC2 from CHILP and ILC2p, and this cell generation was accompanied by an obvious increase in ST2 protein levels on these cells. Consistently, TGF-β significantly upregulated *Il1rl1* mRNA in both CHILP and ILC2p cells. This finding encouraged us to propose ST2 signaling as one of the reasonable downstream mechanisms for ILC2 development programmed by TGF-β signaling. However, some studies have argued that IL-33/ST2 pathway is dispensable for ILC2p development[5,52], although ST2-deficient mice show defect of ILC2 in the periphery[52]. Thus, the exact function of ST2 as a downstream molecule of TGF-β signaling in ILC2 precursors remains to be elucidated.

Thirdly, TGF-β-induced ST2 induction in ILC2 progenitors is at least partially mediated by MEK1/2-dependent, but TAK1- and Smad-independent, pathways. Although the canonical Smad-dependent pathway is known to be required by TGF-β signaling-mediated regulation of many immune cells, our data revealed that TGF-β signaling-mediated regulation of ILC2 may depend on a noncanonical pathway. Supporting this conclusion is the fact that ILC2s and ILC2p have similar to upregulate ST2 in response to TGF-β1 stimulation in the absence of *Smad3*. In TGF-β-associated non-Smad pathways, the TGF-β receptors transmits signal through other factors, such as by the Ras-extracellular signal-regulated kinase (Ras-MEK-ERK) pathway, or the TAK1-c-Jun N-terminal kinase (JNK)-p38-MAPK pathways[33]. Our results showed that the Ras-MEK-ERK pathway plays a major role in TGF-β-mediated upregulation of ST2 in ILC2p and CHILP cells. The Ras-MEK-ERK pathway has recently been demonstrated to play an important role in the maintenance and function of ILC2s[53]. Although the detailed molecular mechanisms underlying TGF-β-mediated-regulation of ST2 remain to be elucidated, our data suggest that this might not rely on the GATA3. It is possible that there are other mechanisms that are linked to ST2 expression as well as ILC2p-ILC2 development programmed by TGF-β signaling. For example, both BCL11b and RORα have been demonstrated as critical transcription factors for ILC2 development[36,54]. Although our data showed that the expression of both *Bcl11b* and *Rora* gene were decreased in both *Tgfbr2*$^{-/-}$ ILC2p and ILC2 cells albeit without statistical difference, *Bcl11b* and *Rora* mRNA were slightly upregulated in WT CHILP cells upon TGF-β1 treatment in cultures.

Fourthly, TGF-β signaling exerts positive effects on mature ILC2s via enhancing the expression of ST2. IL-33 acts through ST2 in promoting the expansion, activation, type-2 cytokine expression, and the migratory capacity of ILC2s[43,44,52]. The regulation of ST2 expression and IL-33-ST2 pathway is critical for the late stage of ILC2 differentiation and functional maturation[51]. The findings that the maintenance, proliferation, and cytokine production (e.g. IL-5, IL-13), as well as respiratory allergic response, of mature ILC2s were impaired and were accompanied by reduced expression of ST2 in the absence of TGF-β signaling suggests a role of TGF-β in the homeostasis and function of mature ILC2s. Consistent with this idea, TGF-β1 treatment enhanced ST2 levels in mature ILC2s. Our study is in agreement with another study[24] showing that mice lacking TGF-β specifically in the bronchial epithelium displayed a reduction in ILC2s

numbers in the lungs and a reduction in IL-13 expression. The study provided additional mechanism that epithelial TGF-β is acting as a chemoactive factor for ILC2s, by enhancing the basal migration of airway ILC2s[24].

In current paradigm, it is generally accepted that both TβR1 and TβR2 are essential components for mediating TGF-β signaling, and deletion of either one could abrogate TGF-β signal transduction[55,56]. For example, T-cell-specific deletion of TβR2[16] or TβR1[14] resulted in lethal inflammation in mice with similar abnormal phenotype and function of T cells. Although we have shown in our most recently published paper that the expression levels (changes) of TβR1, but not TβR2, reflect TGF-β signaling in T cells, both of them are required for TGF-β signaling[57]. Thus, we used *Tgfbr2* deletion in this study. As both blockade of TGF-β signaling with the TβR1 specific inhibitor in cultures and *Tgfbr1*$^{-/-}$ CHILP cultures produced expected results, we expect that the similar results would be obtained with the use of *Tgfbr1*-deleted mice. Interestingly, a recent study showed that the deficiency of Smad4 promotes the conversion of NK into ILC1s-like cells and impairs the function of differentiated NK cells via a TβR1-dependent but TβR2-independent pathway, suggesting a differential roles of TβR1 vs. TβR2 in ILC1/NK cells[26]. Whether TβR1 and TβR2 play different roles in ILC2 development remains an intriguing question that awaits further investigations.

Taken altogether, we have revealed roles of TGF-β in the development of ILC2p and ILC2, and the maintenance of mature ILC2 function with regulation of ST2 expression. Here we propose a model for the ILC2 development programmed by TGF-β (Supplementary Fig. 11): TGF-β signaling is essential for the induction of ST2 from CHILP and enhances the development and generation of ST2$^+$ ILC2p from CHILP; once ST2$^+$ ILC2p are generated, TGF-β signaling is needed to maintain the levels of ST2 expression and prevent ST2 downregulation in ILC2p; in addition, TGF-β signaling also supports the maintenance and function of ILC2 cells in the periphery by regulating their ST2 expression.

## Methods

**Mice.** C57BL/6 and CD45.1 mice were obtained from the Jackson Laboratory. *Tgfbr2*$^{f/f}$ER-Cre$^+$ mice (on the C57BL/6 background), *Smad3*$^{-/-}$ mice, *Tak1*$^{f/f}$ER-Cre$^+$ mice, and *Rag2*$^{-/-}$*Il2rg*$^{-/-}$ mice were bred under specific pathogen-free conditions in the animal facility of the National Institute of Dental and Craniofacial Research. All animal studies were performed according to National Institutes of Health guidelines for use and care of live animals and were approved by the Animal Care & Use Committee (ACUC) of the National Institute of Dental and Craniofacial Research. We have complied with all relevant ethical regulations for animal testing and research.

**Isolation of cells**. Mice were euthanized and systemically perfused by injection of PBS into the left heart ventricle. BM single-cell suspensions were prepared from femurs, tibias, and humerus by flushing the shaft with buffer using a syringe and a 25 G needle. A lineage cell depletion kit (Miltenyi Biotec Inc., 130-090-858) was used for enrichment of lineage-negative cells. Whole lungs were minced and digested in DMEM (10% FBS) containing 1 mg/ml collagenase type IV (Gibco) and 0.5 mg/ml DNase I (Roche Diagnostics) for 20 min at 37 °C with shaking at 100 rpm. The liver was thoroughly dissected and gently passed through a 200-gauge stainless-steel mesh. Small intestines and colons were removed and opened longitudinally and were washed in PBS to remove contents. Intestines were cut into segments 2 cm in length, then transferred into pre-warmed IEL isolation media (DMEM, 4% FBS, 5 mM EDTA, and 0.145 mg/ml DTT) and incubated for 20 min at 37 °C with stirring. The remaining intestinal tissue was washed three times with DMEM containing 2 mM EDTA to remove epithelial cells and fat tissue, and then was minced and incubated in DMEM digestion medium containing liberase (10 mg/ml) and 0.05% DNase for 30 min at 37 °C with gentle shaking. Digested suspensions were filtered and washed in PBS. Lymphocytes from lung, liver, and gut were all isolated using 30% Percoll solution in DMEM. Cells were then filtered with a 70-μm cell strainer and red blood cells were lysed.

**Antibodies for flow cytometric analysis**. All antibodies used for flow cytometry (from eBioscience, BioLegend, or BD Biosciences) are listed in Supplementary Table 2 (clone designations in parentheses).

**Flow cytometry**. For flow cytometry staining, all samples were pre-incubated with an Fc receptor blocking antibody against CD16/32. For ILC staining, lineage-positive cells were excluded using specific antibodies such as: anti-mouse CD3, anti-mouse CD5, anti-mouse CD19, anti-mouse B220, anti-mouse CD11b, anti-mouse CD11c, anti-mouse Gr-1, and anti-mouse CD23. For intracellular staining of transcription factors, cells were permeabilized using FOXP3/Transcription Factor Staining Buffer Set (eBioscience) and stained with specific monoclonal anti-mouse RORγt, anti-mouse GATA3, anti-mouse T-bet or anti-mouse Ki67 antibodies. For intracellular cytokine staining, cells were stimulated for 4 h at 37 °C with PMA (phorbol 12-myristate 13-acetate; 5 ng/ml), ionomycin (1 µg/ml), and Golgi-Plug (1:1000 dilution; BD Pharmingen), followed by staining with fixation/permeabilization buffer solution according to the manufacturer's protocol (BD Biosciences). Flow cytometric analysis was performed on an LSRFortessa instrument and data were analyzed using FlowJo 10 software. For some in vitro and in vivo experiments, cells were sorted using FACSAria III instrument (BD Biosciences).

**Animal treatment**. To generate $Tgfbr2^{-/-}$ mice, 8-12-week-old $Tgfbr2^{f/f}$ER-Cre$^+$ mice were treated with tamoxifen (Sigma-Aldrich) at 1 mg/d i.p. for five consecutive days vs. sunflower oil alone. One day after the last treatment, mice were euthanized. A total of $2–5 \times 10^4$ mature $Tgfbr2^{-/-}$ or WT ILC2s (CD45.2$^+$Lin$^-$CD127$^+$Sca1$^+$CD25$^+$KLRG1$^+$) were sorted from the LP of $Tgfbr2^{f/f}$ER-Cre$^+$ mice that had been treated with tamoxifen or oil, and then were i.v. injected into $Rag2^{-/-}Il2rg^{-/-}$ mice (CD45.2) and euthanized 12 days later.

**Generation of BM chimeric mice**. A total of $2–5 \times 10^6$ BM cells from tamoxifen or oil-treated $Tgfbr2^{f/f}$ER-Cre$^+$ mice were mixed at a 1:1 ratio with BM cells from age- and gender-matched CD45.1 mice, and were i.v. injected into lethally irradiated age- and gender-matched CD45.1 recipients. Mice were analyzed 6 weeks later. Alternatively, BM cells from $Tgfbr2^{f/f}$ER-Cre$^+$ mice without any treatment mixed with CD45.1 BM cells at a 1:1 ratio were transferred into lethally irradiated CD45.1 mice, and these recipients were immediately given an intraperitoneal injection once daily of 1 mg tamoxifen or sunflower oil alone for five consecutive days. Six weeks later, the mice were analyzed. Highly purified $4 \times 10^3$ CD45.2$^+$Lin$^-$CD127$^+$Flt3$^-$ α$_4$β$_7^+$ ILC-potential progenitors from BM of $Tgfbr2^{f/f}$ER-Cre$^+$ mice were mixed with $2 \times 10^6$ whole CD45.1 BM cells and then adoptively transferred into lethally irradiated CD45.1 mice. These recipients were then immediately treated with tamoxifen 1 mg/d i.p. for five consecutive days or oil alone. Six weeks later, CD45.2$^+$ donor-derived ILCs were analyzed.

**In vitro differentiation assays**. Whole BM was extracted as described above. Lin$^-$CD127$^+$Flt3$^-$α$_4$β$_7^+$CD25$^-$ CHILP or Lin$^-$CD127$^+$Flt3$^-$α$_4$β$_7^+$CD25$^+$ ILCp2 cells were sorted from BM of WT C57BL/6, tamoxifen-treated $Tgfbr2^{f/f}$ER-Cre$^+$ or $Tgfbr1^{f/f}$ER-Cre$^+$ mice, $Smad3^{-/-}$, or $Tak1^{f/f}$ER-Cre$^+$ mice. Following isolation, the cells were immediately transferred to confluent 1500 rad X-ray-irradiated OP9-DL1 in 96-well plates, at a density of $0.5 \times 10^3$ to $3 \times 10^3$ cells per well in complete RPMI 1640 (containing 10% fetal bovine serum, 50 µM 2-mercaptoethanol, 1% penicillin–streptomycin, 1 mM sodium pyruvate, 1× non-essential amino acids, and 20 mM HEPES, pH 7.4) with the addition of IL-7 (20 ng/ml) and IL-33 (20 ng/ml) (BioLegend) for ILC2-polarizing conditions, with or without supplementation of rhTGF-β1 (2 ng/ml), or with addition of SB431542 (5 µM) or anti-TGF-β antibody (50 µg/ml). For deletion TAK1, CHILP and ILC2p cells were purified from $Tak1^{f/f}$ER-Cre$^+$ mice and pretreated with 4-hydroxytamoxifen (15 µM) in vitro for 24 h. Cells were passaged and then were added on monolayers of fresh OP9-DL1 cells every 6 days. Cells were harvested for FACS analysis 13 days later. For quantitative RT-PCR analysis, purified CHILP and ILC2p cells were cultured with OP9-DL1 cells in the aforementioned ILC2-polarizing medium, with or without TGF-β1, or with the addition of SB431542 (5 µM), or the TAK1 inhibitor 5z-7oxozeaenol (50 nM), or the MEK inhibitor U0126 (10 µM) for 24 h. CD45$^+$Lin$^-$CD127$^+$Sca$^-$1$^+$CD25$^+$KIRG1$^+$ILC2s were sorted from the LP of WT C57BL/6 mice and cultured in complete RPMI 1640 medium in the presence of IL-7 (20 ng/ml) alone or with the addition of TGF-β1 (0.02-2 ng/ml) or SB431542 (5 µM) for 24 h. Cells were then analyzed by FACS or quantitative RT-PCR.

**Quantitative RT-PCR**. Total RNA was derived from sorted or cultured cells with an RNeasy Plus Micro Kit (QIAGEN), and cDNA was synthesized using a High Capacity cDNA Reverse Transcription Kit (Applied Biosystems). Quantitative real-time PCR was performed according to the protocol of the TaqMan gene expression assay kits (Applied Biosystems) with the following primers: *Hprt*, Mm00446968_m1; *Tgfbr1*, Mm00436964_m1; *Tgfbr2*, Mm03024091_m1; *Gata3*, Mm00484683_m1; *Il1rl1*, Mm00516117_m1; *Gfi1*, Mm00515853_m1; *Bcl11b*, Mm00480516_m1; *Rora*, Mm01173766_m1; *Ets1*, Mm01175819_m1; *Tcf7*, Mm00493445_m1; and *Sox4*, Mm00486320_s1. Results were calculated using comparative ΔΔCT method.

**RNA-seq analysis**. LP CD45$^+$Lin$^-$CD127$^+$Sca$^-$1$^+$CD25$^+$KIRG1$^+$ ILC2 cells were sorted from tamoxifen or oil-treated $Tgfbr2^{f/f}$ER-Cre$^+$ mice. BM CD45.2$^+$Lin$^-$CD127$^+$α$_4$β$_7^+$CD25$^+$ ILC2p cells were sorted from $Tgfbr2^{-/-}$/45.1 or Control/45.1 chimeras. RNA was extracted using RNeasy Plus Micro Kit (QIAGEN), and RNA-seq experiments were performed. Sequence reads were mapped to the mouse genome (mm9) using tophat (with multiple hit = 15)[58]. To quantify the mRNA level of genes, the RKPM (reads per kilobase of transcript per million mapped reads) measures were computed over the exon regions of genes for each gene from the RNA-seq datasets. Differentially expressed genes between the treated and control cells were identified by edgeR[59] using two replicates in each condition (FDR < 0.1 and fold-change > 1.5). For the heatmap in Fig. 3 and Supplementarty Fig. 4, biological relevant genes were selected from literatures, and the q-values between the two conditions were calculated by edgeR, and NS stands for non-significant (q-value > 0.2).

**HDM-induced lung inflammation**. WT and $Tgfbr2^{-/-}$ ILC2s were sorted from the LP, lung, and mesenteric lymph nodes of $Tgfbr2^{f/f}$ER-Cre$^+$ mice that had been treated with tamoxifen or oil, and then were transferred into $Rag2^{-/-}Il2rg^{-/-}$ mice separately ($5 \times 10^5$ per mouse). Twelve days later, recipient $Rag2^{-/-}Il2rg^{-/-}$ mice were injected intranasally with 100 µg of HDM in 50 µl of PBS on days 0–2 and further euthanized on day 3. BALF was collected and lungs were dissected for FACS analysis. Lung sections prepared from HDM-treated mice were stained with H&E stain.

**Statistical analysis**. Groups were compared with GraphPad Prism 6 by a two-tailed unpaired Student's t-test or one-way analysis of variance with post-hoc Bonferroni's test. A p value of <0.05 was considered significant. No data have been excluded from statistical analysis.

**Reporting summary**. Further information on research design is available in the Nature Research Reporting Summary linked to this Article.

## Data availability
RNA-seq data that support the findings of this study have been deposited in NCBI geo database with the accession code GSE140168.

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

## Acknowledgements

This research was supported by the Intramural Research Programs of NIDCR, NILHB, NIH, and the National Key Project for Research & Development of China (No. 2016YFA0502204). We thank the FACS core of NIDCR for their technical assistance. The current author affiliation of J.T. has been renamed to The 901st Hospital of the Joint Logistics Support Force of PLA.

## Author contributions

L.W., J.T., X.Y., and P.Z. designed and carried out the experiments, analyzed the data, and drafted the manuscript. K.C., W.L.K., W.J., D.Z., N.G., and A.C. designed and/or performed the experiments; B.N., K.Z., and Y.W. designed the experiments and supervised the research. W.J.C. conceived of and supervised the research, designed the experiments, and wrote the paper.

## Competing interests

The authors declare no competing interests.
