## [Peer Review File · Nature Communications]

Reviewers' comments:

Reviewer #4 (Remarks to the Author):

The authors have sufficiently addressed my previous concerns

Reviewer #5 (Remarks to the Author):

In this manuscript, Wang et al. provide evidence that TGFbeta is involved in the development of ILC2 in the bone marrow, via its effect on the expression of the *Il1rl1* gene encoding the IL-33R component ST2. The manuscript reflects an impressive body of work and includes many technically challenging experiments. In general, the presented data are convincing and the manuscript reads well. I feel that the authors have largely addressed the concerns raised by the referees.

Nevertheless, it remains unclear how the phenotype of TGFbetaR2-deficient ILC2 relates to the phenotype described for mice lacking IL-33R signaling, in which normal development of ILC2s was reported, but ILC2s were retained in the bone marrow via augmented expression of CXCR4 (Stier et al., *J Exp Med.* 2018). So, it appears that the defects in the absence of TGFbeta signalling cannot only be attributed to reduced IL-33R expression. This is not discussed and CXCR4 expression was not analysed.

- The results presented in the Figures 5 and 6E lack statistical evaluation.
- Regarding the analysis of HDM-mediated allergic airway inflammation (Fig. 7), the authors only show analyses of HDM-treated mice. As a result, it remains unclear how e.g. the numbers of eosinophils or IL-5- or IL-13 producing ILC2 compare to those in unchallenged mice.

Point-by-point Response to Reviewers' Comments and Questions

Reviewer #4 (Remarks to the Author):

The authors have sufficiently addressed my previous concerns

We thank the reviewer for his/her positive comments and support.

Reviewer #5 (Remarks to the Author):

In this manuscript, Wang et al. provide evidence that TGF-beta is involved in the development of ILC2 in the bone marrow, via its effect on the expression of the *Il1rl1* gene encoding the IL-33R component ST2. The manuscript reflects an impressive body of work and includes many technically challenging experiments. In general, the presented data are convincing and the manuscript reads well. I feel that the authors have largely addressed the concerns raised by the referees.

We thank the Reviewer for his/her positive comments on our study. We have addressed the reviewer's remaining questions one by one as follows:

- (1) It remains unclear how the phenotype of TGFbetaR2-deficient ILC2 relates to the phenotype described for mice lacking IL-33R signaling, in which normal development of ILC2s was reported, but ILC2s were retained in the bone marrow via augmented expression of CXCR4 (Stier et al., *J Exp Med.* 2018). So, it appears that the defects in the absence of TGFbeta signalling cannot only be attributed to reduced IL-33R expression. This is not discussed and CXCR4 expression was not analysed.

The comments are valid and well taken. As we responded to the similar comments from the Reviewer #4 in our last response letter, we have in this study provided compelling evidence that TGF- β signaling plays a critical and intrinsic role in the development of ILC2 and ILC2p in the bone marrows. Importantly, we have discovered that the induction of ST2 on CHILP, as well as the maintenance of ST2 expression on ILC2p and ILC2 is most significantly regulated by TGF- β signaling. As previously published studies suggested that IL-33/ST2 signaling was important for ILC2 development and maturation (*Nature immunology* 2012, 13: 229-236; *Nature immunology* 2013, 14(12): 1229-1236; *Immunity* 2018, 48(2): 258-270 e255), we proposed the change of ST2 expression as one of reasonable downstream mechanisms for ILC2 development programmed by TGF- β signaling. However, as this reviewer mentioned, other studies have argued that IL-33/ST2 pathway is not necessary for ILC2p development (*Nature*, 2010, 464: 1367-1370 ; *J Exp Med*, 2018, 215(1):263-281). Based on the controversial role of ST2 in the ILC2 development, it is possible that the ILC2p developmental phenotypes in ST2-knockout mice might not be exactly same as the effects of ST2 deficiency on ILC2p development in T β R2^{-/-} bone marrow-chemaric mice in our paper. We thus would agree with the reviewer that the defects in the absence of TGF- β signaling cannot only be attributed to reduced IL-33R expression. In line with this argument, our data

showed that the expression of *Bcl11b* and *Rora* genes (both have been reported to be important in ILC2 development: *J. Exp. Med.* 2015, 212: 865–874; *Nat Immunol*, Wong et al., 2012) were decreased in both $T\beta R2^{-/-}$ ILC2p cells and $T\beta R2^{-/-}$ ILC2 cells albeit without statistical difference (**Supplementary Fig. 5**), and *Bcl11b* and *Rora* mRNA were also upregulated in WT CHILP cells upon TGF- β 1 treatment *in vitro* cultures (**Supplementary Fig. 7** and data not shown). Thus, we could speculate that TGF- β signaling-mediated development of ILC2s might also be regulated through upregulation of other genes such as *Bcl11b* and *Rora* in CHILP, which might be linked to (or independent on) the up-regulation of ST2 expression. An exciting question need to be elucidated in future studies, but beyond the scope of our current paper. Nevertheless, we discussed these possibilities in our revise manuscript (page 25, 26, 27).

In response to the reviewer’s comment on CXCR4 expression, we have cited this reference (Stier et al., *J Exp Med.* 2018) in our revised manuscript, and have also examined the expression of CXCR4 in the $T\beta R2^{-/-}$ ILC2ps. Firstly, our RNAseq data show that both freshly isolated $T\beta R2^{-/-}$ ILC2p and $T\beta R2^{-/-}$ ILC2s, which already showed reduced ST2 expression, did not show significant changes of the levels of *Cxcr4* mRNA (data not shown). To further study this, we performed more experiments. We cultured WT or $T\beta R2^{-/-}$ ILC2p with OP9-dl1 feeder cells for 6 days and examined the expression of *cxcr4* mRNA (**Reviewer Fig. 1** below). We noticed that the expression of *cxcr4* mRNA was higher in $T\beta R2^{-/-}$ ILC2p compared to WT ILC2p. This is consistent with the phenotype of ST2-deficient ILC2ps with higher levels of CXCR4, but clearly cannot explained the deficiency of bone marrow ILC2p in the $T\beta R2^{-/-}$ bone marrow-chemaric mice. Nevertheless, we have added some discussion about CXCR4 to reflect the reviewer’s point in the text (page 25, 26)

Reviewer fig.1 Quantitative RT-PCR analysis of *cxcr4* gene expression in WT & $T\beta R2^{-/-}$ ILC2p . Data are representative of 2 independent experiments and are presented as mean \pm SD.

(2) The results presented in the Figures 5 and 6E lack statistical evaluation.

We thank for the Reviewer's point. We have now added the statistical evaluations in the Figures 5 and 6E. (revised **Fig. 5 and 6E**).

(3) Regarding the analysis of HDM-mediated allergic airway inflammation (Fig. 7), the authors only show analyses of HDM-treated mice. As a result, it remains unclear how e.g. the numbers of eosinophils or IL-5- or IL-13 producing ILC2 compare to those in unchallenged mice.

We appreciate the reviewer's insightful comment. In fact, we did try to do this type of experiment, but unfortunately failed to isolate sufficient number of cells from the lungs of unchallenged Rag2^{-/-}IL2rg^{-/-} mice that had been transferred with WT or TβR2^{-/-}-ILC2s for a meaningful analysis because of the extremely technical challenges. This was why we could only show the data from the gut of the unchallenged Rag2^{-/-}IL2rg^{-/-} mice (Fig. 6). Only in the HDM challenge, could we have sufficient number of cells to investigate the intrinsic impact of TβR2 deficiency on the function of ILC2 cells response in the lungs (Fig. 7). Importantly, we have demonstrated that the ability to produce IL-5 and IL-13 upon HDM-stimulation was significantly reduced at single cell level in TβR2^{-/-} ILC2s when compare to that in WT ILC2s (Fig. 7). Nevertheless, we have added a statement to reflect this point of missing data in the unchallenged mice (Page 21). We hope that we have answered the reviewer's question.

REVIEWERS' COMMENTS:

Reviewer #5 (Remarks to the Author):

The authors have adequately addressed my comments and carefully revised the discussion section of the manuscript. I do not have any further concerns.

Response to the Reviewer's comments:

Reviewer #5 (Remarks to the Author):

The authors have adequately addressed my comments and carefully revised the discussion section of the manuscript. I do not have any further concerns.

We thank the reviewer for her/his positive comments.